# Childhood Cardiovascular Health, Obesity, and Some Related Disorders: Insights into Chronic Inflammation and Oxidative Stress

**DOI:** 10.3390/ijms25179706

**Published:** 2024-09-07

**Authors:** Tjaša Hertiš Petek, Nataša Marčun Varda

**Affiliations:** 1Department of Paediatrics, University Medical Centre Maribor, Ljubljanska ulica 5, 2000 Maribor, Slovenia; tjasa.hertispetek@ukc-mb.si; 2Faculty of Medicine, University of Maribor, Taborska ulica 8, 2000 Maribor, Slovenia

**Keywords:** cardiovascular health, obesity, obesity-related comorbidities, chronic inflammation, oxidative stress, child

## Abstract

Childhood obesity and associated metabolic abnormalities have become pressing public health concerns worldwide, significantly impacting cardiovascular health. Metabolic syndrome, characterized by a cluster of metabolic abnormalities including central obesity, altered glucose metabolism, dyslipidemia, and arterial hypertension, has emerged as a critical precursor to cardiovascular disease. Chronic systemic inflammation and oxidative stress seem to play pivotal roles in the pathogenesis of childhood obesity-related disorders such as early atherosclerosis. A significant distinction between the objective components of cardiovascular health metrics, including body mass index, blood pressure, cholesterol, and fasting glucose levels, and the definition of metabolic syndrome is evident in the identification of obesity. Whereas cardiovascular health metrics predominantly rely on body mass index percentiles to assess obesity, metabolic syndrome criteria prioritize waist circumference, specifically targeting individuals with a measurement ≥90th percentile. This discrepancy emphasizes the need for a nuanced approach in assessing the risks associated with obesity and underscores the importance of considering multiple factors when evaluating cardiovascular risk in children. By recognizing the complex interplay between various health metrics, obesity and metabolic syndrome criteria, clinicians can more accurately identify individuals at risk and tailor interventions accordingly to mitigate cardiovascular disease in children with obesity.

## 1. Introduction

Cardiovascular diseases (CVDs) are the number one morbidity and mortality group of diseases in developed society and include coronary, cerebrovascular, aortic, and peripheral vascular disorders [1]. The origins of cardiovascular disease are at the beginning of life [2]. Pathological data have shown that atherosclerosis begins in childhood and that the extent of atherosclerotic change in pediatric populations can be correlated with the presence of the same risk factors identified in adults [3].

There are different definitions for cardiovascular health. Cardiovascular health, as defined by the American Heart Association for adults, comprises seven health metrics (smoking, body mass index, physical activity, diet, total cholesterol, blood pressure, and fasting plasma glucose) [4]. Recently, they added duration of sleep [5]. Another good way to evaluate cardiovascular risk in pediatric populations is by identifying individuals with metabolic syndrome (MetS), which represents a group of interrelated risk factors that predict CVD and diabetes mellitus (DM) [6].

Furthermore, childhood pre-obesity/overweight and obesity have been found to be positively correlated with both chronic systemic inflammation and oxidative stress [7,8]. There has been an increasing interest in exploring the connections between chronic inflammation, oxidative stress, childhood cardiovascular health in relation to obesity, and associated disorders, topics that are covered in this review article.

## 2. Cardiovascular Health Metrics

For clear and consistent monitoring and messaging throughout childhood, age-appropriate scoring of the cardiovascular health components based upon the American Heart Association’s “cardiovascular health” construct has been provided. It includes eight metrics: diet, physical activity and screen time, sleep, smoking exposure, body mass index (BMI), blood pressure, cholesterol, and glucose measurements [2].

Alcohol consumption, although not directly included in current metrics, is a significant risk factor for the future development of cardiovascular disease. Evidence suggests that even low levels of smoking and alcohol use are associated with increased arterial stiffness. Therefore, public health strategies should focus on preventing these behaviors during adolescence to maintain cardiovascular health throughout the lifespan [9,10].

It is reasonable to initiate healthful lifestyle training in childhood to promote improved cardiovascular health in adulthood [3]. Some national guidelines recommend evaluation for cardiovascular risk factors such as obesity and arterial hypertension as part of general pediatric care [2].

For preventive measures, primary care counseling will mostly be focused on the four cardiovascular behaviors (diet, physical activity, sleep, and smoking exposure), whereas children with poor levels of cardiovascular factors (BMI, blood pressure, cholesterol, and glucose) may sometimes require referral for intensive management in secondary or tertiary health institutions [2]. It is also recommended that health and education professionals, child health advocates, policymakers, and community leaders are informed about the metrics and become interested in optimizing the school environment as an integral part of population-based strategies designed to promote cardiovascular health in pediatric populations and reduce the risk and public health burden of cardiovascular diseases [11].

## 3. Cardiovascular Diseases

Some of the most important risk factors for the development of cardiovascular diseases are obesity and arterial hypertension. Large cohort studies have, thus, demonstrated that high blood pressure is an important risk factor for heart failure, atrial fibrillation, chronic kidney disease, heart valve diseases, aortic syndromes, and dementia, in addition to coronary heart disease and stroke [12].

The common denominator, atherosclerosis, is a pathologic process defined as being, among other definitions, chronic inflammation. The inflammatory process is established through endothelial cell activation [1]. Endothelial cells are both the source and target of numerous factors contributing to atherosclerosis [13]. In fact, several hundred variables were shown to be associated with coronary disease [1]. The presence of an oxidized low-density lipoprotein and its deposition inside the arterial wall, recognition by macrophages, and subsequent proinflammatory immune response is thought to be a major pathogenic mechanism of the atherosclerotic cascade [14]. In healthy children, systemic chronic inflammation and oxidative stress already influencing cardiovascular health seem to take place, which was demonstrated by a positive correlation between circulating inflammatory chemokines and oxidative stress markers with vascular characteristics of the carotid artery [15,16].

## 4. Chronic Systemic Inflammation and Oxidative Stress in Cardiovascular Diseases

Oxidative stress and inflammation are closely related pathophysiological processes, one of which can be easily induced by another. Thus, both processes are simultaneously found in many pathological conditions [17]. Chronic inflammation is referred to as slow, long-term inflammation lasting for prolonged periods of several months to years. Generally, the extent and effects of chronic inflammation vary with the cause of the injury and the ability of the body to repair and overcome the damage [18]. Cytokines are known to be involved in several inflammation-related processes and must be regulated properly. However, their expression, production, or activity are affected by several genetic and environmental factors [1,19]. Oxidative stress is defined by an imbalance between the production and generation of reactive oxygen species (ROS) in cells and tissues and the capability of an organism to scavenge these molecules via antioxidant mechanisms, some of which are presented in Table 1 [20]. Because of the importance of this topic, several biomarkers of chronic inflammation and oxidative stress in connection to cardiovascular health and disease were investigated (Table 1) [20].

## 5. Metabolic Syndrome and Obesity Markers in Children

Another good way to evaluate cardiovascular risk in pediatric populations is by identifying children and adolescents with MetS. MetS represents a group of interrelated risk factors that predict cardiovascular diseases and diabetes mellitus [6]. People with MetS have higher mortality rate when compared with people without MetS, primarily caused by progressive atherosclerosis, accelerated by pro-inflammatory state, oxidative stress, and pro-coagulation components of MetS [6,21].

Even though there is no consensus on a MetS definition for children and adolescents, most authors agree on essential components such as glucose intolerance, central obesity, arterial hypertension, and dyslipidemia, each representing a risk for cardiovascular disease [22]. The definition of MetS in children according to the International Diabetes Federation (IDF) [23] applies to children aged 10 to 16 years and refers to waist circumference ≥ 90th percentile (ethnic-specific waist circumference [24]) AND number of abnormalities ≥ 2: Triglyceride ≥ 150 mg/dL (1.7 mmol/L), HDL < 40 mg/dL (1.03 mmol/L), BP either: systolic > 130 mmHg or diastolic ≥ 85 mmHg, and fasting glucose ≥ 100 mg/dL (5.6 mmol/L) [23]. For children 16 years and older, the adult criteria can be used. In comparison to adults’ definition of MetS, the definition for children aged 10 to 16 years uses ethnic-specific waist circumference percentiles and one cutoff level for high-density lipoprotein (HDL) rather than a sex-specific cut-off. For children younger than 10 years of age, MetS cannot be diagnosed. However, caution is still recommended if the waist circumference is ≥90th percentile or other metabolic abnormalities are noticed [25,26,27].

The main difference between objective components of cardiovascular health metrics (BMI, blood pressure, cholesterol, and glucose) and MetS (IDF definition) is the identification of obesity. In cardiovascular health metrics, they use BMI percentiles and in MetS, they use waist circumference ≥90th percentile (Table 2). In the following passage, we describe MetS and its individual components and objective criteria of cardiovascular health in the light of systemic chronic inflammation and oxidative stress.

## 6. Metabolic Syndrome, Chronic Systemic Inflammation, and Oxidative Stress

MetS in children has been linked to a heightened risk of cardiovascular disease and all-cause mortality in adulthood [28]. Waist circumference measurement, a crucial anthropometric parameter in children with obesity and MetS, can be easily obtained during a physical examination and may serve as a predictor of increased cardiovascular risk [29].

The etiology of MetS is multifactorial, involving both genetic predisposition and environmental risk factors. Individuals with MetS often have family members affected by its components, indicating a genetic influence, as well as an unhealthy lifestyle in the particular family. Additionally, the presence of MetS in parents can predict subclinical inflammation in children, potentially contributing to the development of atherosclerotic disease in the future [30].

An essential characteristic of MetS, alongside altered glucose metabolism leading to an elevated risk of developing type 2 DM [31], dyslipidemia, and arterial hypertension, is abdominal obesity [8,32]. Excess weight, particularly concentrated in the abdominal area, is linked to heightened secretion of adipokines and inflammatory cytokines [33]. Specifically, visceral fat is recognized as more than just a passive energy storage depot. It functions as an active endocrine organ, secreting various bioactive molecules. In obesity, the overproduction of proinflammatory and prothrombotic adipokines contributes to inflammation, which plays a central role in the pathogenesis of MetS and its impact on CVD [34].

Key adipokines in MetS encompass a range of biomarkers such as interleukins (including interleukin-1, -6, -10, and -18), adiponectin; resistin; tumor necrosis factor alpha; leptin; monocyte chemoattractant protein-1; angiotensinogen; plasminogen activator-inhibitor-1; myeloperoxidase (MPO) and E-selectin [34,35]. Newer biomarkers also include carotenes, tocopherols, and dietary vitamins with antioxidant properties, all of which were found to be lower in metabolically unhealthy children [36].

The coexisting presence of MetS and elevated inflammation markers has been associated with a more substantial increase in arterial stiffness and carotid intima-media thickness, both of which are measures of vascular changes in atherosclerosis [37]. Ceruloplasmin and 8-isoprostane have shown potential as useful indicators for identifying patients at a heightened risk of future CVD [38,39].

Bilirubin may also contribute to the development of MetS. In addition to its roles in the biliary and hematologic systems, bilirubin possesses potent antioxidant and cytoprotective properties, enabling it to inhibit various stages of atherosclerosis formation. Studies have linked bilirubin levels with carotid intima-media thickness and ischemic cardiac disease. Serum total bilirubin levels have also shown an inverse correlation with the prevalence of MetS, with the mechanism of this association possibly linked to insulin resistance [40].

In Table 3, we present some oxidative stress and inflammation markers in obesity and related disorders.

### 6.1. Glucose Intolerance, Chronic Systemic Inflammation, and Oxidative Stress

DM type 2 is a prevalent comorbidity of obesity and MetS. Over the past few decades, there has been a notable rise in the disease burden among children. Understanding the underlying pathways contributing to the inflammatory and vascular complications associated with these conditions is a key focus in this field. In this context, irisin emerges as a potential player in the pathophysiology. Irisin, a recently discovered adipomyokine, is primarily secreted by skeletal muscles in response to acute exercise, with some contribution from adipose tissue. Serum irisin levels were found to be significantly reduced in children with MetS or DM type 2, showing negative correlations with BMI percentile. Consequently, decreased irisin levels may lead to reduced inhibition of oxidative stress and inflammation [41].

Various other adipokines, including leptin, adiponectin, resistin, and visfatin, are secreted by adipose tissue and are elevated in obesity, contributing to the pathophysiology of obesity-related disorders [42], particularly in the development of DM type 2 [43]. Leptin, the most well-known adipokine, has been predictive of interleukin-6 (IL-6) levels in young individuals and is associated with the degree of overweight. Elevated IL-6 levels have been observed in young children not only with obesity but also with fully developed MetS [44]. Furthermore, IL-6 levels were found to be elevated in children with DM type 2 [43]. Additionally, stromal-derived factor and soluble E-selectin are potential indicators of the onset of insulin resistance and endothelial damage [29].

Interestingly, research on adults with obesity has revealed that acute glucose consumption leads to a temporary decline in endothelial function and an increase in inflammation and oxidative stress, a phenomenon that was not observed in youth with obesity. Nonetheless, this association may hold implications for individuals with impaired glucose tolerance or DM type 2 [45].

### 6.2. Dyslipidemia, Chronic Systemic Inflammation, and Oxidative Stress

Dyslipidemias refer to quantitative alterations in total cholesterol concentration, individual fractions, or triglycerides within the plasma. Research indicates that dyslipidemia during childhood is linked to the development of atherosclerosis in adulthood [46]. Early detection and intervention have the potential to mitigate cardiovascular risk later in life, as cardiovascular disease remains the primary cause of morbidity and mortality in developed nations [47].

Oxidative stress and chronic systemic inflammation might play an important role in children with hypercholesterolemia. Oxidative stress could potentially play a substantial role in children with obesity-related hypercholesterolemia. This is evidenced by elevated levels of nicotinamide-adenine dinucleotide phosphate oxidase and oxidized low-density lipoprotein (LDL) compared to groups of healthy children, children with obesity alone, or children with familial hypercholesterolemia. The correlation between multiple cardiovascular risk factors and nicotinamide-adenine dinucleotide phosphate oxidase is associated with increased endothelial dysfunction and heightened oxidative stress in children [48,49]. Even asymptomatic patients with moderate and severe hypercholesterolemia exhibit evidence of oxidant stress, as demonstrated by lipid peroxidation characterized by increased levels of F2 isoprostanes [50]. An important mediator in the formation of the atheroma plaque is myeloperoxidase (MPO). It plays a significant role by promoting LDL oxidation and making HDL dysfunctional [51,52].

Furthermore, inflammatory and hemostatic abnormalities were observed in children with familial hypercholesterolemia. They included elevated levels of plasminogen activator-inhibitor-1, interleukin-1β, intracellular cell adhesion molecules, and impaired endothelial function as indicated by endothelium-dependent reactive hyperemia and endothelium-independent nitrate hyperemia dilatation [53]. Children with familial hypercholesterolemia also demonstrated notably higher gene expression of chemokines, with elevated levels of RANTES (Regulated upon Activation, Normal T Cell Expressed and Presumably Secreted), a chemokine and chemoattractant for monocytes, memory T-helper cells, and eosinophils. Namely, inflammation in hypercholesterolemia seems to be modulated by monocyte-derived RANTES [54]. These findings suggest an inflammatory pathophysiological basis for endothelial dysfunction and atherosclerosis in children with elevated levels of cholesterol [53].

### 6.3. Arterial Hypertension, Chronic Systemic Inflammation, and Oxidative Stress

Another characteristic of MetS is arterial hypertension (AH) [8,32]. In children, the diagnostic criterion for AH is based on the normal distribution of blood pressure (BP) in healthy children. Blood pressure levels are defined by the child’s age, gender, and height. The 2016 European guidelines for assessing blood pressure in children and adolescents use blood pressure tables derived from the 2004 American guidelines [55,56]. In adults, the definition of AH is based on numerical values, while in pediatrics, as indicated above, it relies on percentile curves. The exception are adolescents over the age of 16, when the cut-off values for elevated systolic and diastolic blood pressures are the same as for adults (140/90 mm Hg) according to European guidelines. In children under the age of 16, normal blood pressure is defined as systolic or diastolic blood pressure below the 90th percentile, high-normal BP is ≥90th to <95th percentile, stage 1 AH is 95th–99th percentile + 5 mm Hg, and stage 2 AH is >99th percentile + 5 mm Hg. Isolated systolic hypertension is diagnosed when systolic BP is ≥95th percentile and diastolic BP is below the 90th percentile [55,57].

Essential AH, characterized by a blend of genetic predisposition and environmental factors, plays a role in accelerating atherosclerosis and cardiovascular diseases [58]. In recent years, elevated blood pressure has been on the rise, primarily due to the increasing prevalence of overweight and obesity, which is one of the main risk factors [59,60,61]. The latest data on the prevalence of elevated blood pressure among children and adolescents in Slovenia were published in 2018, estimating it to be around 3–4% [61]. As a result, the rise of obesity-related arterial hypertension has become a noteworthy cardiovascular concern in children, stemming from various factors associated with obesity. These include alterations in endocrine functions, such as corticosteroids and adipokines, increased activity of the sympathetic nervous system, disturbances in sodium balance, and the presence of oxidative stress, inflammation, and endothelial dysfunction [62].

The origins of AH may even trace back to the prenatal period, where exposure to adverse in utero conditions leads to the generation of reactive oxygen species. This, in turn, triggers the developmental programming of AH [58]. However, the exact timing of vascular alterations remains uncertain, and discerning whether these differences arise from genetic factors or perinatal influences can be challenging [63].

In healthy neonates with a significant family history of myocardial infarction, the expression of inflammation-related molecules in endothelial cells was notably heightened compared to neonates without such familial predisposition. Offspring of hypercholesterolemic mothers displayed distinct arterial gene expressions, suggesting potential genetic programming in utero [63]. Overweight and obesity during pregnancy have been linked to higher birth weights, childhood obesity, and noncommunicable diseases in offspring [64]. Similarly, ultrasound examinations of the neonatal and fetal aorta suggest that impaired fetal growth, maternal hypercholesterolemia during pregnancy, and diabetic macrosomia constitute significant risk factors for vascular changes indicative of early-stage atherosclerosis [65].

In line with the fetal programming hypothesis, neonates exposed to placental insufficiency due to malnourishment exhibit endothelial cell dysfunction. The impairment of nitric oxide production or activity appears to be a fundamental factor in endothelial dysfunction, a process exacerbated by mitochondrial damage in malnourished fetuses [66]. Moreover, fetal hypoxia induces the generation of reactive oxygen species, leading to oxidative stress, which exacerbates endothelial cell dysfunction and promotes vascular smooth muscle cell proliferation and apoptosis [66,67]. To conclude, being small for gestational age also poses a risk factor for the development of type 2 DM, MetS, and CVD later in life [66].

The repercussions of disrupted fetal programming resulting from an adverse prenatal environment have been observed in children before puberty, with elevated inflammation markers being evident [52]. This cardiovascular risk emerges early in life, as evidenced by boys aged 6–8 years who are considered at risk (based on maternal cardiovascular health and lifestyle habits), exhibiting heightened markers of oxidative stress, arterial stiffness, and elevated diastolic blood pressure [68].

Genetic predisposition appears to significantly influence the development of cardiovascular risk as well, as evidenced by ethnic disparities. Variations in the prevalence of risk factors and diseases across racial and ethnic groups are attributed to the frequency of particular genotypes and their interactions with environmental factors. Notable differences include elevated blood pressure, lower triglyceride levels, and higher levels of HDL lipoprotein cholesterol observed in individuals of Black ethnicity. Hispanics exhibit a higher prevalence of DM and insulin resistance, while Japanese individuals tend to have elevated triglyceride levels compared to Caucasians [69]. However, the precise contributions of prenatal, genetic, and postnatal environmental and behavioral factors remain to be fully understood [64].

Low-grade chronic inflammation is implicated in the development of essential AH, with vascular inflammation preceding systemic inflammatory alterations [70]. Intriguingly, systemic oxidative stress has been documented in hypertensive children and adolescents, also independent of their BMI, characterized by decreased nitrate levels and increased lipid peroxidation end products. The ratio between lipid peroxidation and nitric oxide levels correlates directly with both systolic and diastolic blood pressures across the entire patient population. Also, they exhibit reduced antioxidative capacity due to significant glutathione depletion compared to matched controls based on BMI [71].

As indicated above, children with increased cardiovascular risk have impaired antioxidants/oxidative stress balance such as reduced endogenous levels of vitamins C and E, reduced glutathione, increased levels of malondialdehyde, and oxidized low-density lipoproteins as well as decreased levels of superoxide dismutase (SOD) and decreased total antioxidant capacity [72]. Research is currently delving into antioxidant mechanisms, such as thiol/disulfide homeostasis. Thiols are pivotal in safeguarding cells against oxidative stress. When oxidative stress occurs, thiols transform into reversible disulfide structures, reverting to thiol groups once the stress subsides. Elevated disulfide levels indicate increased oxidative stress, as evidenced in adolescents with essential AH [73]. Additionally, urinary biomarkers of oxidative stress have demonstrated correlation with an arteriosclerosis index in school children, hinting at their potential to non-invasively predict lifestyle-related disease risks [74].

### 6.4. Obesity, Chronic Systemic Inflammation, and Oxidative Stress

Children who present with BMI ≥ 5th and <85th percentile for age and sex are considered to have a normal weight, those with BMI ≥ 85th percentile for age and sex are considered to be in an overweight range, and those with BMI ≥ 95th percentile for age and sex are considered to have obesity [75,76].

In recent decades, obesity has surged to epidemic levels among children and is also linked to the emergence of various other cardiovascular risk factors [20].

Although some complications of obesity in children and adolescents are less common than in adults, we can expect them to significantly increase morbidity and mortality in the entire population in the long run [77]. Monitoring children and adolescents with overweight in the context of cardiovascular health, chronic inflammation, and oxidative stress is important, as addressing excess weight, one of the primary factors for cardiovascular disease development, can be most effectively prevented through early interventions in childhood [59,60,61].

Oxidative stress and inflammation play a significant role in the development and progression of CVD in obesity. Increased production of reactive oxygen species (ROS) and decreased antioxidant defense mechanisms lead to oxidative damage to lipids, proteins, and nucleic acids [20]. A state of oxidative stress is believed to be linked to the release of adipocytokines, which are associated with obesity as well as with MetS and the activation of the renin-angiotensin-aldosterone system. Research suggests that oxidative stress is associated with organ damage, such as left ventricular hypertrophy and carotid intima-media thickness, in hypertensive children, in addition to metabolic abnormalities, adiposity, and insulin resistance [78]. Furthermore, obesity is thought to be related to changes in adipokine secretion, activity of adipose tissue macrophages, helper T cells, and regulatory 4T cells [79] as well as with adipogenesis, lipid metabolism, and thermogenesis [80].

Interferon-γ–alpha T-cell chemoattractant (I-TAC/CXCL11) is a novel biomarker of inflammation in obesity, where an association between endothelial leukocyte stasis and increased risk of cardiovascular morbidity in obesity was suggested. Additionally, it is associated with adipose tissue angiogenesis in adults with obesity [81,82].

Adiponectin is nathe most abundant serum adipokine, mainly secreted from white adipose tissue, and has demonstrated antiatherogenic, anti-inflammatory, and insulin-sensitizing effects. In obese children, decreased levels of adiponectin have been documented, with plasma levels showing an inverse correlation with abdominal obesity [83]. Adiponectin plasma levels were also lower in children with type 2 DM [43]. Furthermore, adiponectin change was found to be positively related to a change in flow-mediated dilation and negatively to change in arterial stiffness (incremental elastic modulus) and pulse wave velocity [84]. In patients with carotid intima-media thickness > 2 SD, the level of adiponectin was lower than in those with normal one [70]. Low levels of adiponectin were associated with the development of cardiovascular complications of obesity and were associated with CVD disease even in children and adolescents [83].

Monocyte chemoattractant protein-1 (MCP-1) is a chemokine that is upregulated by pro-inflammatory macrophage pathways, thereby facilitating the infiltration of pro-inflammatory immune cells into adipose tissue. MCP-1 was significantly increased in obese children. There exists a notable inverse relationship between MCP-1 levels and insulin sensitivity [85,86]. It also plays an important role in AH in adults [87]; however, a study on children failed to show this correlation [70].

A recent systematic review identified an inverse association between vitamin D status and biomarkers of oxidative stress and inflammation, including C-reactive protein, interleukin-6, cathepsin S, vascular cell adhesion molecule-1, malondialdehyde, MPO, 3-nitrotyrosine, and SOD. However, this association was observed in only five out of eight studies reviewed. Furthermore, some of these studies employed adjusted models to account for obesity, including BMI and body fat thickness [88].

Some other biomarkers were also investigated in children with obesity such as chemerin, which is connected to increased inflammation endothelial activation [89,90], and elevated systolic pressure [91]; catestatin, which is significantly lowered in obesity and connected to the reduction of adrenergic stimulation and regulation of oxidative stress [92]; and insulin-like growth factor binding protein-3, which is reduced in obesity and is believed to be an early marker of atherosclerosis [93]. Interestingly, the total white blood cells count and neutrophil parameters alone were found to have a positive predictive value in estimating the degree of cardiovascular risk among the individuals with obesity [94].

Elevated homocysteine levels have also been linked to increased cardiovascular risk [95]. In children with obesity, hyperhomocysteinemia is associated with disrupted homeostatic regulation, leading to elevated levels of proinflammatory chemokines. These chemokines are believed to play a role in the early stages of the inflammatory process associated with atherosclerosis [96].

In assessing oxidative stress, investigations are underway to measure protein and lipid oxidation products for potential clinical applications [97]. Studies have shown elevated concentrations of oxidized LDL in children with obesity even before an increase in carotid intima-media thickness occurs [98]. Moreover, elevated levels of urine 8-isoprostane have been associated with BMI, waist circumference, and ambulatory blood pressure [99].

MPO is a peroxidase enzyme produced in neutrophils and monocytes involved in host defense. It produces oxidizing compounds such as hypochlorite from hydrogen peroxide [52]. It is increased in children and adolescents with obesity [52,100,101]. Elevated levels of MPO are associated with increased risk of CVD [52]. MPO plays a crucial role in atherogenesis and is related to endothelial damage in incipient stages of the process [52,102,103]. Specifically, it has been linked to cardiovascular risk through mechanisms, including the release of reactive oxygen species, oxidation of LDL particles, conversion of HDL into dysfunctional particles, endothelial dysfunction, platelet aggregation, and vulnerability of atheromatous plaques [51,52].

Additionally, research is exploring antioxidants like thiol/disulfide homeostasis, which showed impairment in obesity, suggesting its involvement in oxidative stress and inflammation associated with obesity [104]. Nitric oxide is also a focus of investigation, with its levels increased in correlation with fat accumulation and leading to elevated values of cardiometabolic risk markers in children [105]. Polyamines, originating from arginine (a precursor of nitric oxide), are under scrutiny and have been found to be notably higher in children with obesity [106].

SOD is a key antioxidant enzyme, which is activated in cells to fight against oxidative stress [107,108,109] and was found to be decreased in individuals with obesity [108]. There are three isoforms of SOD in human: the cytoplasmic Cu/ZnSOD (SOD1), the mitochondrial MnSOD (SOD2), and the extracellular Cu/ZnSOD (SOD3). Each isoform requires a catalytic metal (Cu or Mn) for activation [109]. Specifically, SOD serves as the primary antioxidant defense systems against excessive reactive oxygen species, particularly by partitioning the superoxide anion (O_2_•−) into normal molecular oxygen (O_2_) and hydrogen peroxide (H_2_O_2_) [108,109]. Additionally, it is thought that excessive reactive oxygen species play significant roles in the pathogenesis of various CVDs, including AH and atherosclerosis [109,110]. Hence, the reduction in SOD levels observed in obesity could potentially contribute to the heightened cardiovascular burden associated with this condition.

Exploring the proteomic profile of peripheral blood mononuclear cells (PBMCs) offers a complementary approach to assessing key biological markers of oxidative stress and chronic inflammation, presenting a promising avenue for investigation. PBMCs, readily isolated from peripheral blood, primarily consist of circulating lymphocytes (comprising 70–90% of cells), monocytes (10–20%), and dendritic cells. Recent research on adults and children has demonstrated that proteomic patterns could potentially serve as early indicators of metabolic changes, offering opportunities for the early detection and potential treatment of obesity-related conditions [111,112,113]. Analyzing PBMC protein expression in children and adolescents is anticipated to provide valuable insights into the status of immune cells, which may vary between those with normal weight and those with overweight/obesity.

To conclude, obesity and its related conditions contribute to the progression of atherosclerosis, affecting not only adults but also children [13]. It is clear that systemic inflammation and oxidative stress impact the cardiovascular health of children and young adults from an early age [114], emphasizing the need for further detailed research in this area (Figure 1).

## 7. Essential Recommendations for Clinicians

Early intervention and consistent monitoring are cornerstones in preventing cardiovascular disease. Family history of premature cardiovascular disease, arterial hypertension, dyslipidemia (especially elevated LDL cholesterol), diabetes mellitus type 2, and tobacco exposure should be assessed. Children with a family history of myocardial infarction, angina pectoris, coronary artery bypass graft/stent/angioplasty procedures, and sudden cardiac death in the first-degree relative at <55 years for males and <65 years for females are specially at higher risk and require early and more frequent monitoring to mitigate the development of cardiovascular conditions [2,115,116]. Furthermore, early intervention is also needed in children with conditions like chronic kidney disease, congenital heart defects, chronic inflammatory diseases, and those who have undergone organ transplants or cancer treatments [117].

Clinicians should encourage healthy dietary habits, physical activity, and the management of risk factors such as lipid levels, blood pressure, and body weight for all children. A diet abundant in adequate intakes of fruits and vegetables, fiber-rich whole grains, fish, and limited intakes of sugar-sweetened beverages and sodium, all scaled for caloric intake, is essential for maintaining optimal cardiovascular function. Exclusive breastfeeding during the first six months of life is recommended since it is associated with long-term cardiovascular benefits [2,116].

Physical activity plays a vital role in cardiovascular health, with at least 60 min of moderate to vigorous exercise daily recommended for children and adolescents. Sedentary time should be limited to 2 h per day. Preschool children should be offered opportunities for active play daily. Regular physical activity enhances cardiovascular fitness, helps maintain a healthy body weight, and reduces the risk of developing obesity-related conditions [2,115,116].

Tobacco and alcohol use are significant modifiable risk factors for cardiovascular disease. The early initiation of tobacco and alcohol use has profound implications for cardiovascular health, necessitating strong efforts to create and maintain smoke-free environments and to prevent the initiation of tobacco use and alcohol consumption among adolescents. Furthermore, E-cigarette use is rising and poses risks to cardiovascular health and may lead to nicotine addiction. Pediatric providers should promote smoke-free environments, offer strong anti-smoking messages, and provide resources for smoking cessation [2,9,10,115,116].

Routine screening of cholesterol is essential for the early detection of dyslipidemia, a key risk factor for atherosclerosis. Universal lipid screening is recommended once before the onset of puberty and again in young adulthood. These age intervals are important for identifying children at risk and for initiating appropriate interventions before the development of more serious cardiovascular conditions [115,116].

Routine blood pressure monitoring should begin at the age of 3 years and be conducted regularly thereafter. Early detection of arterial hypertension is essential, as untreated high blood pressure in childhood can lead to significant cardiovascular complications later in life [116].

Body mass index is a critical measure for assessing obesity, a major risk factor for cardiovascular disease. In addition to BMI, waist circumference serves as an important indicator of metabolic syndrome, a cluster of conditions that increase the risk of cardiovascular disease and type 2 diabetes mellitus [116].

Pharmacological interventions may be necessary when lifestyle modifications are insufficient to manage cardiovascular risk factors. For children with persistent arterial hypertension unresponsive to lifestyle changes, antihypertensive medications such as angiotensin converting enzyme inhibitors, angiotensin receptor blockers, or calcium channel blockers may be used. In children over 10 years old with severe hyperlipidemia, statin therapy may be considered, particularly when lifestyle interventions alone do not adequately lower cholesterol levels. Monitoring non-HDL cholesterol, alongside LDL and total cholesterol, is advised to manage dyslipidemia effectively. In managing type 2 diabetes mellitus, lifestyle interventions are the first line of defense. However, when glycemic control cannot be achieved through lifestyle changes alone, pharmacological treatments such as metformin or insulin are introduced to manage blood sugar levels effectively [115,116,117].

Furthermore, emerging evidence in the therapy of obesity highlights the efficacy of newer pharmacological agents, specifically GLP-1 receptor agonists (GLP-1RAs), which significantly reduce body weight, BMI, and waist circumference in patients without diabetes mellitus. Among these agents, semaglutide has shown superior outcomes with a favorable safety profile in adults as well as in adolescents, marked by fewer gastrointestinal side effects compared to liraglutide and exenatide. These findings underscore the potential of GLP-1RAs as powerful tools in obesity management [118,119].

Regarding inflammation and oxidative stress state, which are recognized factors in the atherosclerotic process, the role of inflammatory markers as independent risk factors in children has been considered. Although inflammation was included as a potential independent risk factor in the evidence review, the National Heart, Lung, and Blood Institute Expert panel did not find sufficient evidence to support the measurement of inflammatory markers in children in routine clinical practice. Only a small number of randomized controlled trials and observational studies included measurements of selected inflammatory markers, and these were not sufficient to make any conclusive statements for measuring inflammatory markers in the pediatric age group. Therefore, while inflammation is critical in adult atherosclerosis, its role in pediatric cardiovascular health remains unclear, and no recommendations for routine measurement of these markers in children have been made [120]. This highlights the need for continued research into the roles of inflammation and oxidative stress in pediatric cardiovascular health, addressing gaps in current evidence. Our review identifies these critical areas, supporting future studies and better-informed treatment strategies. Meanwhile, all the described activities, namely, primordial, primary, secondary, tertiary, and quaternary prevention, have to be promoted to provide optimal cardiovascular health [121].

## 8. Conclusions

Obesity and its related conditions contribute significantly to the development of atherosclerosis, not only in adults but also in children. Children and adolescents who are overweight or with obesity need to be monitored regarding their cardiovascular health, chronic inflammation, and oxidative stress. Addressing excess weight, a primary factor in cardiovascular disease development, is most effectively achieved through early interventions during childhood. Further research is needed to elucidate the complex interplay between metabolic syndrome, obesity, systemic inflammation, oxidative stress, and cardiovascular health.

## Figures and Tables

**Figure 1 ijms-25-09706-f001:**
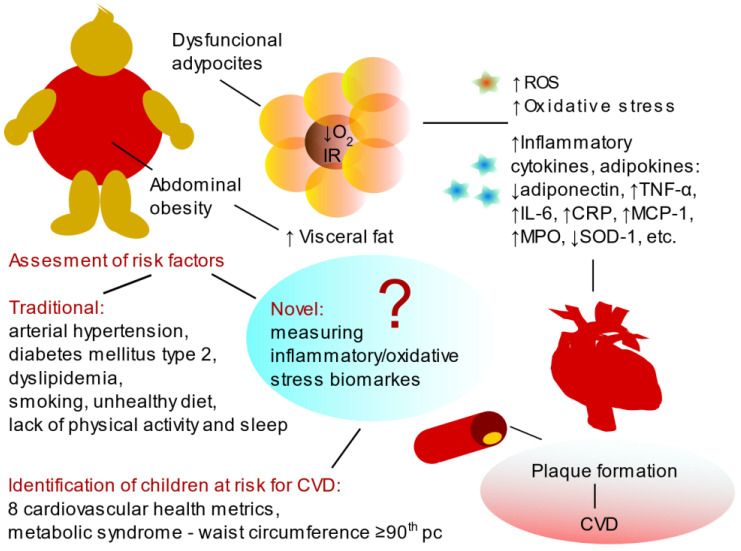
Abdominal obesity, characterized by increased visceral fat, is closely linked to the secretion of inflammatory adipokines and cytokines that induce peripheral insulin resistance (IR). The hypoxic environment (↓O2) within hypertrophic, dysfunctional adipocytes promotes the generation of reactive oxygen species (ROS), driving oxidative stress and perpetuating both local and systemic low-grade chronic inflammation. This inflammatory state is pivotal in the development of atherosclerotic plaques, a key factor in cardiovascular disease (CVD). While traditional risk factors remain central to cardiovascular health assessment, an emerging approach may include biomarkers of inflammation and oxidative stress. However, expert consensus is required to validate their clinical application. Tumor necrosis factor-alpha (TNF-α), interleukin-6 (IL-6), C-reactive protein (CRP), monocyte chemoattractant protein-1 (MCP-1), and myeloperoxidase (MPO).

**Table 1 ijms-25-09706-t001:** Oxidative and inflammatory markers associated with cardiovascular disease and some antioxidant markers.

Oxidative Stress Markers	Antioxidant System Markers	Adipokines and Other Systemic Inflammation Markers
Lipid peroxidation F2 isoprostanes malondialdehyde (MDA) thiobarbituric acid reactive substances (TBARSs) oxidized low-density lipoprotein (oxLDL) Protein oxidation advanced oxidation protein products (AOPPs) Carbohydrate oxidation advanced glycosylation end-products (AGEs) Nucleic acid oxidation 8-hydroxy-2′-deoxyguanosine (8-OHdG) Reactive oxygen species (ROS) generation myeloperoxidase (MPO) NADPH ^1^ oxidase (NOX2) Nitric oxide system (NOx) polyamines derived from arginine asymmetric dimethyloarginine (ADMA) nitrite and nitrate NO	thiol/disulphide homeostasis glutathione (GSH) superoxide dismutase (SOD) catalase (CAT) glutathione peroxidase (GPx) carotenes (vitamin A) ascorbic acid (vitamin C) tocopherols (vitamins E) bilirubin ceruloplasmin Total antioxidant capacity (TAC)	chemerin adiponectin leptin resistin visfatin adipomyokine irisin RANTES ^2^ monocyte chemoattractant protein−1 (MCP−1) stromal-derived factor (SDF−1) interleukins IL-1, −1β, −6, −10, −18 tumor necrosis factor alpha (TNF−α) plasminogen activator-inhibitor−1 (PAI−1) α-1-acid glycoprotein (AGP)high sensitivity C-reactive protein (hsCRP) C-reactive protein (CRP) myeloperoxidase (MPO)

^1^ NADPH, nicotinamide-adenine dinucleotide phosphate oxidase. ^2^ RANTES, Regulated upon Activation, Normal T Cell Expressed and Presumably Secreted.

**Table 2 ijms-25-09706-t002:** Comparing the International Diabetes Federation definition of metabolic syndrome with objective criteria of cardiovascular health metrics for children.

Criteria	Metabolic Syndrome in Children (IDF ^1^)	Cardiovascular Health Metric for Children
Waist circumference	≥90th percentile for age and sex	Not specifically included
Triglycerides	≥1.7 mmol/L (150 mg/dL)	Elevated triglycerides (cutoffs vary by age)
High-density lipoprotein cholesterol	<1.03 mmol/L (40 mg/dL) in males <1.29 mmol/L (50 mg/dL) in females	Low HDL cholesterol (cutoffs vary by age and sex)
Blood pressure	Systolic BP ≥ 130 mmHg or diastolic BP ≥ 85 mmHg	≥90th percentile for age, sex, and height
Fasting glucose	≥5.6 mmol/L (100 mg/dL)	≥5.6 mmol/L (100 mg/dL)
Body mass index	Not specifically included	Within normal range for age and sex (≥5th and <85th percentile)
Total cholesterol	Not specifically included	Acceptable range (varies by age)

^1^ IDF—International Diabetes Federation.

**Table 3 ijms-25-09706-t003:** Some oxidative stress and inflammation markers in obesity and related disorders.

Condition	Some Oxidative Stress and Inflammation Markers
Metabolic Syndrome	Interleukins (IL-1, IL-6, IL-10, IL-18), adiponectin, resistin, tumor necrosis factor alpha, leptin, monocyte chemoattractant protein-1, angiotensinogen, plasminogen activator-inhibitor-1, myeloperoxidase, e-selectin, carotenes, tocopherols, some other dietary vitamins
Glucose Intolerance	Irisin, leptin, adiponectin, resistin, visfatin, stromal-derived factor, soluble e-selectin
Dyslipidemia	Nicotinamide-adenine dinucleotide phosphate oxidase, oxidized low-density lipoprotein, plasminogen activator-inhibitor-1, interleukin-1β, intracellular cell adhesion molecules, RANTES ^1^
Arterial Hypertension	Nitrate levels, lipid peroxidation end products, glutathione depletion, oxidized low-density lipoprotein, myeloperoxidase, urine 8-isoprostane, antioxidative capacity, thiol/disulfide homeostasis, nitric oxide, superoxide dismutase
Obesity	Reactive oxygen species, adipocytokines, I-TAC/CXCL11 ^2^, adiponectin, monocyte chemoattractant protein-1, chemerin, catestatin, homocysteine, superoxide dismutase, nitric oxide, polyamines, thiol/disulfide homeostasis, myeloperoxidase

^1^ RANTES—Regulated upon Activation, Normal T Cell Expressed and Presumably Secreted, ^2^ I-TAC/CXCL11—Interferon-γ–alpha T-cell chemoattractant

## Data Availability

No new data were created or analyzed in this study. Data sharing is not applicable to this article.

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
