# Peer review of "Childhood Cardiovascular Health, Obesity, and Some Related Disorders: Insights into Chronic Inflammation and Oxidative Stress"

_ijms, 2024, doi:10.3390/ijms25179706_

Round 1

Reviewer 1 Report

Comments and Suggestions for Authors

The review was very clearly written and addresses an important emerging area of interest.   I have a few suggestions to add to the review:

1.  Alcohol consumption is a risk factor for adults.  Given onset of drinking in teens/early adults, should it be considered in the review as a risk factor for this population?

2. The review was well written but a summary figure or two would help break it up and clearly show the author's main points.

Author Response

Comments 1:The review was very clearly written and addresses an important emerging area of interest.   I have a few suggestions to add to the review:

  1. Alcohol consumption is a risk factor for adults.  Given onset of drinking in teens/early adults, should it be considered in the review as a risk factor for this population?

Response 1: Thank you for pointing this out. We have added a paragraph on alcohol consumption as a cardiovascular risk factor on page 2, second paragraph, lines 53 to 57. Additionally, we mention alcohol consumption on page 12, fourth paragraph, lines 481 to 488. 

Comments 2: The review was well written but a summary figure or two would help break it up and clearly show the author's main points.

Response 2: We agree with this comment, therefore we have added a summery figure on page 11.

Reviewer 2 Report

Comments and Suggestions for Authors

This review provides an excellent coverage of the complicated are of childhood obesity and its consequences. It covers all areas very thoroughly but does not summarise or conclude with guidelines for the strongest evidence for the prevention or treatment of childhood cardiovascular health. It is an excellent literature review but needs to present where the strong recommendations are to be followed to minimise the onset of the problems and guide physician treatment options

Author Response

Comments 1: This review provides an excellent coverage of the complicated are of childhood obesity and its consequences. It covers all areas very thoroughly but does not summarise or conclude with guidelines for the strongest evidence for the prevention or treatment of childhood cardiovascular health. It is an excellent literature review but needs to present where the strong recommendations are to be followed to minimise the onset of the problems and guide physician treatment options

Response 1: Thank you for pointing this out. We have added a new section, page 11 to 13, lines 456 to 536, which addresses this issue.